# Multimodal Functionalities of HIV-1 Integrase

**DOI:** 10.3390/v14050926

**Published:** 2022-04-28

**Authors:** Alan N. Engelman, Mamuka Kvaratskhelia

**Affiliations:** 1Department of Cancer Immunology and Virology, Dana-Farber Cancer Institute, Boston, MA 02215, USA; 2Department of Medicine, Harvard Medical School, Boston, MA 02115, USA; 3Division of Infectious Diseases, Anschutz Medical Campus, University of Colorado School of Medicine, Aurora, CO 80045, USA

**Keywords:** HIV, integrase, virus maturation, virus morphogenesis, allosteric integrase inhibitor, integrase-RNA binding, aberrant integrase multimerization

## Abstract

Integrase is the retroviral protein responsible for integrating reverse transcripts into cellular genomes. Co-packaged with viral RNA and reverse transcriptase into capsid-encased viral cores, human immunodeficiency virus 1 (HIV-1) integrase has long been implicated in reverse transcription and virion maturation. However, the underlying mechanisms of integrase in these non-catalytic-related viral replication steps have remained elusive. Recent results have shown that integrase binds genomic RNA in virions, and that mutational or pharmacological disruption of integrase-RNA binding yields eccentric virion particles with ribonucleoprotein complexes situated outside of the capsid shell. Such viruses are defective for reverse transcription due to preferential loss of integrase and viral RNA from infected target cells. Parallel research has revealed defective integrase-RNA binding and eccentric particle formation as common features of class II integrase mutant viruses, a phenotypic grouping of viruses that display defects at steps beyond integration. In light of these new findings, we propose three new subclasses of class II mutant viruses (a, b, and c), all of which are defective for integrase-RNA binding and particle morphogenesis, but differ based on distinct underlying mechanisms exhibited by the associated integrase mutant proteins. We also assess how these findings inform the role of integrase in HIV-1 particle maturation.

## 1. Introduction

Retroviruses comprise a comparatively large virus family (*Retroviridae*) that is divided into two subfamilies, Orthoretrovirinae and Spumaretrovirinae, each of which encompasses about a half dozen viral genera. While infections with spumaretroviruses are generally benign, orthoretroviruses can cause serious diseases in animal and human hosts, including cancers (α-, β-, δ-, ε-, and γ-retroviruses) and wasting immunodeficiencies (lentiviruses, which include human immunodeficiency virus (HIV)).

Retroviruses are roughly spherical, lipid-enveloped particles of approximate 90 to 120 nm in diameter. Internal protein-based lattices help to provide structure to the virion particles and play important roles in virus infection. For the spumaretroviruses, such lattices are provided by Gag proteins that are minimally processed by the viral protease (PR) enzyme [1,2]. By contrast, orthoretroviral Gag proteins are proteolyzed into individualized virion structural proteins matrix, nucleocapsid (NC), and capsid (CA). A peripheral hexameric lattice of the HIV-1 matrix interacts directly with the viral lipid bilayer [3]. Internal to the matrix lattice lies the viral core, which consists of a lattice of CA that encases the viral ribonucleoprotein complex (vRNP; Figure 1A). The vRNP encompasses elements essential to viral genome replication, including two copies of plus-stranded viral RNA (vRNA) and viral NC, reverse transcriptase (RT), and integrase (IN) proteins. The HIV-1 capsid lattice, which is a cone-shaped fullerene shell [4], is assembled from approximately 200 CA hexamer and 12 CA pentamer building blocks [5,6].

Virus replication cycles are broadly divided into early versus late temporal steps, and key early events in the lifecycles of orthoretroviruses include cell entry, reverse transcription, and integration (spumaretroviral infection also relies on these steps, but reverse transcription in this case can occur prior to cell entry [8]). During reverse transcription, RT converts vRNA into linear double-stranded viral DNA (vDNA) containing a copy of the long terminal repeat (LTR) at each end [9], which is the substrate for IN-mediated integration [10]. IN functions as an obligate oligomer within the context of the intasome nucleoprotein complex (reviewed in [11] and discussed further below). IN catalyzes two sequential magnesium-dependent S_N_2 transesterification reactions, 3′ processing and strand transfer. During the 3′ processing reaction, IN hydrolyzes vDNA ends adjacent to conserved CA-3′ dinucleotides, exposing reactive vDNA CA_OH_-3′ termini [12,13,14,15]. During strand transfer, IN uses the CA-3′ hydroxyl groups to cut chromosomal DNA in staggered fashion across a major groove, which simultaneously joins the vDNA ends to host DNA 5′-phosphate groups [12,15,16]. Repair of the resulting hemi-integration intermediate, which is accomplished by host cell enzymes, yields integrated proviral DNA flanked by the sequence duplication of the staggered chromosomal DNA cut [11]. Provirus formation marks the transition from the early to the late stage of retroviral replication.

Key late events in the viral lifecycle include transcription and translation to produce vRNA genomes and viral proteins, which are the substrates to assemble new virus particles. In addition to the HIV-1 Gag polyprotein that harbors matrix, NC, CA, and p6 domains, the PR, RT, and IN enzymes are expressed as the C-terminal region of a larger Gag-Pol polyprotein. Ribosome frameshifting of HIV-1 mRNA maintains the stoichiometry of Gag to Gag-Pol synthesis at approximately 20-to-1 [17]. Gag and Gag-Pol proteins together with envelope glycoproteins and vRNA coalesce to bud through the cell plasma membrane to form immature HIV-1 particles (see Ref. [18] for a recent review). Maturation of the virus particle initiates via Gag-Pol dimerization, which activates PR activity to autocatalytically liberate the enzyme from the polyprotein precursor [19,20]. PR can then cleave additional Gag and Gag-Pol polyproteins into individual virion structural proteins and enzymes [21,22]. HIV-1 maturation culminates with the formation of the mature matrix lattice and the viral core [23] (Figure 1A).

RT subunit compositions vary among the retroviruses. In addition to RNA and DNA-dependent DNA polymerase activity, RT enzymes harbor a second active site that catalyzes RNase H (H for hybrid) activity, which degrades the RNA portion of RNA/DNA hybrids during reverse transcription [9]. The RTs of β- [24], δ- [25], and γ-retroviruses [26,27] function as monomers that possess both DNA polymerase and RNase H activities. By contrast, the HIV-1 Pol dimer is asymmetrically processed by PR to yield p66 and p51 RT subunits [28,29]. HIV-1 RT functions as a p66/p51 heterodimer, with the p66 subunit catalyzing both DNA polymerase and RNase H activities [30,31,32]. Alpharetroviral Pol proteins are also asymmetrically processed by PR, in this case to yield heterodimers composed of larger β and smaller α subunits [33,34]. The α subunit harbors both DNA polymerase and RNase H activities [34]. The β subunit is a fusion of RT and IN, and IN was first recognized as a 32 kD protein in avian myeloblastosis virus cores that was related to RT-β and that displayed metal-dependent DNA endonuclease activity in vitro following its purification by conventional chromatography [35].

Virus mutagenesis experiments initially implicated a direct role for IN in vDNA integration [36,37,38,39,40], which was subsequently proven by showing that purified, recombinant IN proteins possessed 3′ processing and DNA strand transfer activities in vitro [41,42,43,44,45,46,47]. Today, enzyme active site inhibitors, which are known as IN strand transfer inhibitors (INSTIs), are important components of antiretroviral therapies that are used to treat people living with HIV [48]. Mutations in IN were additionally recognized early on to in some cases impact HIV-1 replication at steps other than vDNA integration, including virus particle maturation and/or reverse transcription [49,50,51,52,53,54]. More recent work has clarified that IN binds to vRNA in HIV-1 virions and that the disruption of IN-vRNA binding decouples vRNPs from being incorporated into capsid lattices [55,56]. The resulting eccentric virions, so named due to the eccentric location of the vRNP outside of the capsid lattice (Figure 1B), fail to support reverse transcription due to rapid loss of vRNP components from nascently infected cells [56,57]. In this monograph, we will overview the effects of IN mutations on HIV-1 replication and the role of IN in HIV-1 particle maturation. This area of investigation is highlighted by the mechanism of action of a second class of highly potent IN inhibitors, the allosteric IN inhibitors (ALLINIs). ALLINIs induce aberrant IN multimerization, the consequences of which inhibit IN-vRNA binding and accordingly arrest HIV-1 replication at the comparatively late maturation step of vRNP encasement into the protective capsid lattice [7,55,58]. Note, we delineate functional IN oligomerization, which is required for IN assembly onto vDNA ends to form catalytically competent intasomes and high affinity IN-vRNA interactions during virion morphogenesis, from aberrant IN multimerization. Aberrant multimerization, resulting from amino acid substitution (s) or ALLINI treatment, results in non-functional IN through preventing its functional oligomerization.

## 2. Distinct Functions and Structures of HIV-1 IN

IN plays a dual role in HIV-1 biology. During the early steps of infection, IN catalyzes vDNA integration into host chromosomal DNA. IN has a second, non-catalytic role during virion maturation, wherein it interacts with the vRNA genome to ensure the packaging of the vRNP within the viral capsid.

### 2.1. IN Structures and HIV-1 Integration

Appreciation of the role of IN in non-catalytic aspects of the HIV-1 lifecycle necessitates upfront descriptions of its known functions in vDNA integration. The simplified 3′ processing and strand transfer activity assays provided initial platforms to investigate the roles of vDNA nucleotides and protein domains/amino acid residues in IN function. The active site is composed of a catalytic triad of electronegative residues that is phylogenetically conserved among retrotransposon INs and certain bacterial transposase proteins [59,60,61] (the DDE motif [62]). The central region of IN, where the active site resides, was relatively resistant to proteolysis [59], which, together with deletion mutant analyses, led to the definition of three protein domains: the N-terminal domain (NTD), catalytic core domain (CCD), and C-terminal domain (CTD) [63,64,65,66] (Figure 2). The DDE active site coordinates the positions of two magnesium ions that activate attacking nucleophiles (water for 3′ processing; vDNA CA-3′ hydroxyl groups for strand transfer) and destabilize scissile phosphodiester bonds [67]. The NTD harbors His and Cys residues that are conserved across retroviral and retrotransposon INs (the HHCC motif), bind zinc, and are important for IN oligomerization and 3′ processing and DNA strand transfer activities [59,61,64,68,69,70,71]. The CTD, which lacks a virus family-wide conserved amino acid sequence motif, is also critical for IN multimer formation and activity [72,73,74] (Figure 2).

Initial 3D structures were solved for individual HIV-1 IN domains. The NTD is a 3-helical bundle that is stabilized by zinc binding [79]. The CCD is a homodimer wherein each monomer adopts an RNase H fold [80] and the CTD adopts a beta-stranded SH3 fold [81,82]. As expected from amino acid sequence similarities, the secondary structures of the HIV-1 IN domains are conserved among the other retroviral IN proteins (reviewed in Ref. [83]). Diversity among retroviral IN proteins increases in regions outside of the folded globular domains. A flexible ~11-residue linker resides between the HIV-1 IN NTD and CCD domains while the CCD and CTD are separated by an approximate 20-residue linker (Figure 2). Depending on the species of retrovirus, these sequences range in length from about 11 to 20 residues for the NTD-CCD linker and from about 10 to 60 residues for the CCD-CTD linker [78]. An 18-residue “tail” region lies C-terminal to the HIV-1 CTD SH3-fold [77,84] (Figure 2). Across retroviruses, this region varies from just a few residues for lentiviral equine infectious anemia virus IN to as many as ~55 residues for β-retroviral mouse mammary tumor virus IN. Because IN C-terminal tail regions have largely eluded visualization by structural biology techniques, they are seemingly flexible and thus likely to adopt variable conformations.

Experiments conducted in the 1990s first revealed the requirement of oligomerization for IN activity [72,73,85]. However, the detailed interplay between different IN protomers within active complexes awaited the determination of the first retroviral intasome structure, which was solved for the simiispumavirus prototype foamy virus (PFV) [86]. The PFV intasome comprises an IN tetramer that binds to and synapses the vDNA ends together. Although at the time it was generally believed that tetramers were the basic catalytic units of all retroviral INs, subsequent intasome structures revealed a surprising array of protein oligomers among the different complexes. Akin to PFV, δ-retroviral intasomes harbor an IN tetramer [87,88], while α-and β-retroviral intasomes contain higher-order octameric IN arrangements [78,89,90]. Lentiviral intasomes have revealed more diverse structures, which range from IN tetramers to hexadecamers [76,91]; the field consensus is that the higher-order structures likely represent the physiological arrangement of lentiviral IN on vDNA [92,93] (Figure 3).

Regardless of the IN-to-vDNA stoichiometry, all intasome structures solved to date harbor core elements first observed for PFV. The structure that these elements compose is hence called the conserved intasome core (CIC) [91]. Within the CIC, the two IN molecules that provide functional active sites (blue and green in Figure 3) swap their NTDs across the vDNA-bound synaptic interface, which is flanked by rigid, synaptic CTD spacers (colored hotpink in the figure). The blue and green IN molecules are engaged by the CCDs of additional IN protomers that do not engage the vDNA ends (Figure 3, cyan). All IN domains of the inner green-blue IN dimer contact the vDNA ends, while the CCDs and CTDs of these IN protomers additionally bind target DNA for integration [94].

The different IN-to-vDNA stoichiometries across intasome structures in part reflect structural constraints imposed by IN interdomain linker lengths. For example, the comparatively short ~10-residue CCD-CTD linker of α-and β-retroviral INs precludes the positionings of synaptic CTDs that are accomplished in the PFV intasome by much larger 50-mer CCD-CTD linker regions [78,89]. The α-and β-retroviruses accordingly employ additional IN dimers to donate synaptic CTDs to the structures, leading to overall IN octamer contents. Although the lentiviral IN CCD-CTD linkers are ~20 residues, corresponding span lengths are restricted by alpha helical secondary structure. For the lentiviruses, IN tetramers donate the synaptic CTDs, leading to the overall higher ratios of IN-to-vDNA in these structures [76,91].

### 2.2. IN-vRNA Binding during Virion Morphogenesis

Consistent with its affinity for heparin Sepharose resin [45], purified HIV-1 IN protein can effectively bind a variety of negatively charged polymers, including double-stranded DNA [95,96], single-stranded DNA [97], RNA [98], and poly (Asp-Glu) [99]. Given this non-specific binding characteristic, it took several years for the field to hone specific IN-vDNA binding conditions [100,101,102], which critically informed HIV-1 intasome structural studies [76,93]. In terms of IN-RNA binding specificity, key results were determined by crosslinking immunoprecipitation combined with next generation sequencing (CLIP-Seq) using partially purified HIV-1 virions, which revealed reproducible binding of HIV-1 IN to specific regions of the vRNA genome. The pattern of IN-vRNA binding was importantly distinguishable from the NC-vRNA binding pattern, which was also determined by CLIP-Seq [55]. Although IN bound vRNA at numerous positions, the viral transactivation response (TAR) stem-loop element harbored a comparatively high level of sequence reads, and efficient binding of purified HIV-1 IN to synthetic TAR RNA (*K_d_* ~ 3 nM) was demonstrated using an amplified luminescent proximity homogenous assay (tradename AlphaScreen™ [103]). We however note that tight binding of IN to TAR RNA was not detected by a nitrocellulose filter binding assay format [104]. In addition to vRNA binding, the alpha screen assay revealed that IN can effectively bridge two separate TAR RNA molecules together. Consistent with these observations, atomic force microscopy experiments demonstrated that purified IN bridges individual vRNA segments to form condensed ribonucleoprotein structures [55].

HIV-1 IN in solution exists in a concentration-dependent equilibrium between lower order monomer/dimer and higher order tetrameric forms [56,71,72,95,105,106,107,108,109,110,111]. Strikingly, IN tetramers, but neither dimers nor monomers, displayed effective vRNA bridging activities in vitro, indicating that virion-associated IN is predominantly tetrameric [56]. The structural organization of individual IN protomers in the context of IN-vRNA complexes is unknown. We note that different arrangements of IN oligomers, including the domain-swapped vDNA-bound IN dimer and flanking IN tetramers, exist within the HIV-1 intasome (Figure 3). It remains to be seen how individual IN protomers and IN domains are arranged in the complex with vRNA.

Mass spectrometry-based protein footprinting identified three lysine residues in the CTD and C-terminal tail region of HIV-1 IN, Lys264, Lys266, and Lys273, as critical for RNA binding [55] (Figure 4). K264A/K266A and R269A/K273A substitutions in the context of purified recombinant IN fully impaired binding of these mutant proteins to synthetic TAR RNA without affecting functional IN tetramerization. While purified K264A/K266A IN mutant protein was defective for strand transfer activity in vitro, R269A/K273A mutant IN supported the wild type (WT) level of IN strand transfer activity. Despite their efficient incorporation into virions, K264A/K266A and R269A/K273A mutant INs were defective for HIV-1 RNA binding, and the mutant virus particles were predominantly eccentric [55]. Following infection of susceptible target cells, IN mutant R269A/K273A virus was moreover defective for reverse transcription [56,112]. Notably, Lys264, Lys266, Arg269, and Lys273 are positioned along flexible elements of the CTD and adjacent tail region (Figure 4), suggesting a crucial role of this segment of IN for interactions with vRNA [55]. Subsequent mutagenesis experiments identified additional electropositive residues that potentially contact vRNA, including CTD residues Arg262 and Arg263, as well as Lys34, which is located in the NTD. Substitutions of these residues did not compromise IN tetramerization, yet they impaired IN interactions with vRNA in vitro and in virions [56]. While these site-directed mutagenesis studies revealed basic residues which could form electrostatic interactions with vRNA, we suspect that other, yet to be identified IN residues, provide specificity determinants to recognize cognate sequence and/or structural features of vRNA. Consistent with a major contribution from electrostatic interactions, IN-vRNA binding was disrupted by the presence of 1 M NaCl [55].

Initial biochemical assays examined structural features of the TAR stem-loop [113], which revealed the importance of the hairpin loop and adjacent 3 nucleotide bulge for high affinity binding to IN [55]. High-resolution structural studies are necessary to delineate atomic details underpinning specific, high affinity binding of vRNA to IN.

## 3. Phenotypic Spectra of HIV-1 IN Mutant Viruses

Numerous studies have described the outcomes of IN mutagenesis on HIV-1 replication. Initial work revealed the requirement of IN for virus replication in model tissue culture systems, which defined IN as a critical viral component and as such a viable candidate for antiretroviral inhibitor development [40,114]. Although some mutant viral strains were discovered to be specifically blocked at the integration step of the viral lifecycle [114], it soon became apparent that many other mutations impacted additional steps, including particle assembly/release from virus producer cells and/or ensuing reverse transcription in infected target cells. Because IN is expressed as a component of the Gag-Pol polyprotein, it may not, in hindsight, be surprising that some IN mutations could impact viral late events, which rely on the incorporation of the Gag-Pol protein into virus particles and its subsequent dimerization to activate PR-dependent polyprotein processing. Now more than 20 years ago, one of us (Engelman) coined two phenotypic classes to distinguish HIV-1 IN mutant viruses that were specifically blocked at the integration step (class I) from those that display pleiotropic replication defects (class II) [115]. Because class II mutations are by definition pleiotropic, caution should be exercised in interpreting corresponding viral phenotypes, and we indeed believe that underappreciation of HIV-1 IN mutant viral pleiotropy has led to some prior misconceptions. What has become apparent in recent years is that defective IN binding to vRNA in HIV-1 virions is a key determinant of the class II IN mutant viral phenotype [55,56].

Initial IN mutagenesis studies focused primarily on deletion mutant viruses [40,116], which were quickly followed by missense mutagenesis to test the roles of conserved residues such as those that compose metal-binding HHCC and DDE residues [49,50,51,52,54,114,117,118,119,120]. In general, IN deletion and HHCC missense mutant viruses display the class II phenotype. To fully appreciate the myriad of phenotypic changes associated with class II IN mutant viruses and how these inform the role of IN in virus particle maturation, we will first review the class I IN mutant viral phenotype.

### 3.1. Class I IN Mutant Viruses

Typified by changes of the DDE active site residues, class I IN mutant viruses contain the normal complement of virion proteins, harbor a vRNA-binding competent IN, and generally appear phenotypically normal and display the WT level of reverse transcription; in certain studies, subtle reverse transcription (at most 2-fold) and particle morphology defects have also been noted [50,51,52,54,56,75,114,117,118,119,120,121,122,123,124,125]. Class I IN mutant viruses accordingly behave as expected for viruses that are specifically blocked at the integration step of the HIV-1 lifecycle. Class I IN mutant viral infections are in this vein phenotypically similar to WT viral infections conducted in the presence of INSTI inhibitors, with the subtle distinction that DDE active site mutations will prevent IN 3′ processing and strand transfer activities while INSTIs preferentially inhibit IN strand transfer activity [126].

A small percentage of vDNA in infected cell cultures is converted into closed circular forms containing either one LTR, which can form by homologous recombination [127], or two abutted LTRs, which form by non-homologous end joining (NHEJ) [128]. Due to the unique LTR-LTR junction sequence and the fact that NHEJ machinery resides in the nucleus, the 2-LTR circle affords a convenient marker for HIV-1 preintegration complex (PIC) nuclear localization [129]. A hallmark of infection with class I IN mutant viruses is increased levels of 2-LTR circles over those observed in cells infected with WT virus, which is presumably caused by increased availability of linear vDNA for circularization in the absence of functional integration [51,52,75,112,121,122,123,124]. In this vein, the ratio of 2-LTR circles to total vDNA is a key indicator of an integration block, even in cases when vDNA synthesis is low. Such analyses have helped clarify that defective integration is also common among class II IN mutant viruses [84,112,130].

Infection of indicator cell lines that carry integrated reporter genes under the control of the HIV-1 LTR can yield class I IN mutant viral infectivities as high as ~20% of WT, likely due to effective HIV-1 Tat protein expression from unintegrated vDNA [51,118,121,122]. Recent research has clarified that retroviral DNAs are transcriptionally silenced by heterochromatin soon after nuclear entry [131,132,133] and identified cellular proteins that regulate gene expression from unintegrated HIV-1 DNA [134,135]. The δ-retroviral human T lymphotropic virus 1 transcriptional activator Tax protein can also significantly enhance HIV-1 transcription from unintegrated vDNA [136,137]. Research that aims to inform the roles of IN residues or IN inhibitors on HIV-1′s ability to replicate under physiologically relevant conditions might appropriately avoid Tax-expressing cell lines such as MT-4 and C8166-45 [40,138,139,140,141,142,143,144] moving forward.

Alterations of active site proximal residues including Asn120 [52,124], Glu92 [122], Lys159 [75], Gln62, and His67 [124] can elicit the class I IN mutant viral phenotype (Figure 5A). However, in contrast to the DDE catalytic triad residues, not all substitutions of active site proximal residues yield replication-defective mutant viruses. For example, while both E92A and E92Q IN mutant viruses replicated indistinguishably from the WT, E92K was a replication-defective class I IN mutant virus [122]. Similarly, while HIV-1 carrying the IN CTD W235F change replicated similar to the WT [51], replication-defective W235E was phenotypically a class I mutant virus [52]. While DDE active site mutant proteins are defective for IN 3′ processing and strand transfer activities, other class I IN mutant proteins, such as E92K and W235E, can support partial catalytic function in vitro [122,145,146]. PICs extracted from cells infected class I IN mutant viruses, including W235E, failed to support IN strand transfer activity or protect vDNA ends from bacteriophage Mu-mediated DNA footprinting in vitro, indicating defective intasome formation as a common phenotype among class I IN mutant viruses [147].

### 3.2. Class II IN Mutant Viruses

The class II moniker historically encompasses all HIV-1 IN mutant viruses that display replication defects at steps other than integration. All class II IN mutant viruses are defective for IN-vRNA binding, virus particle morphogenesis, reverse transcription, and, to some extent, integration [56,112,130,146]. However, recent research has highlighted that different subclass distinctions within class II can be made. We will first review the common phenotypic features of class II IN mutant viruses and then propose new subclasses based on the distinct underlying mechanistic defects exhibited by the associated IN mutant proteins and viruses.

#### 3.2.1. Adverse Effects on Reverse Transcription

Reverse transcription was first highlighted as a universal defect among class II IN mutant viruses [115]. Viruses containing amino acid substitutions within the IN NTD [51,52,54,56,120,151,152], CCD [49,52,54,56,117,122,124,130,148,153,154,155,156], CCD-CTD linker [56,112,130], or CTD [56,112,123] can display reverse transcription defects, which vary between studies from as little as ~2-fold [155,156] to greater than 1000-fold [154]. Although the reasons for these large differences in reverse transcription defects are not completely clear, they almost certainly rely in part on different methodological approaches. Because replication-defective mutant viruses are produced from cells by DNA transfection, some level of transfected DNA will invariably carry-over to viral stocks, which, following target cell infection, can confound interpretations of reverse transcription levels. Carry-over DNA is commonly degraded by treating virus stocks with a non-specific DNase enzyme. Parallel infections are best conducted in the presence of a potent RT inhibitor to further define bona fide mutant viral reverse transcription levels. As expected, inefficient removal of plasmid DNA from class II IN mutant viral stocks can yield artifactual reverse transcription measures [130].

Reverse transcription assays generally leverage genetically modified single-round viral vector constructs, wherein envelope glycoproteins are supplied in trans by co-transfection. The identity of the envelope glycoprotein has been reported to impact class II IN mutant viral reverse transcription significantly, though in inconsistent ways across studies. While one study reported that vesicular stomatitis virus G (VSV-G) glycoprotein stimulated class II IN mutant reverse transcription ~20- to 90-fold [157], a separate study noted ~2- to 4-fold suppression of IN mutant viral reverse transcription compared to infections initiated via the HIV-1 envelope [156]. Finally, it seems entirely possible that certain class II IN mutations will yield greater reverse transcription defects than others. For example, substitution of Cys130 in the IN CCD with Gly or Ser yielded comparatively robust reverse transcription defects [130,154]. The largest study of class II IN mutant viruses to date, which included 25 missense mutants, reported on average ~10-fold reverse transcription defects, with approximate 4-fold variation (~7% to 30% of WT) across virus samples [56].

IN deletion mutant viruses that lack most of the protein or just the CTD are defective for reverse transcription [51,52,54,118,123,124]. Step-wise deletion of residues from the C-terminus revealed the extent of the IN C-terminal tail region that is required for HIV-1 infection and reverse transcription [84]. In this study, viruses that retained ~10% of WT HIV-1 activity under single-round infection conditions were able to support replicative spread in CEM-SS T cell cultures. While IN mutant 1-273 lacking the C-terminal 15 residues sustained spreading HIV-1 replication, deletion mutant viruses 1-272, 1-271, and 1-270 were replication-defective. These three mutants supported WT profiles of reverse transcription and increased levels of 2-LTR circle formation, and thus were typed as class I mutant viruses. By contrast, mutants deleted beyond position 269 were ~4- to 10-fold defective for reverse transcription and were hence typed as class II IN mutant viruses [84]. These and other data [77] indicate that the IN C-terminal tail region (residues 271–288; Figure 2) is largely dispensable for HIV-1 infection and reverse transcription.

Purified HIV-1 IN and RT proteins can interact directly [54,154,158,159,160], which is mediated via the IN CTD [154,158,159,161]. Moreover, because each protein can stimulate the other’s in vitro activity [159,162], the IN-RT interaction has been proposed to underlie IN’s role in reverse transcription. Surely, at some level, an IN-RT interaction can be envisaged during HIV-1 infection, e.g., as the viral core transitions from the reverse transcription complex [163] to the PIC. However, it is currently unclear if the IN-RT interaction plays a key role in reverse transcription, per se. Consistent with the interpretation that it does, the class II IN K258A CTD mutation [123] reduced binding to RT in vitro [146,161]. At the same time, the class I W235E IN mutation ablated the interaction with RT [146,158]. Thus, interactions of IN mutant proteins with RT in vitro do not necessarily correlate with the efficiency of reverse transcription during IN mutant viral infection [146].

Numerous cellular proteins are packaged into retroviral particles (reviewed in [164] for HIV-1) and IN interactor 1 (INI1)/SMARCB1, which interacts directly with HIV-1 IN [165], is specifically incorporated into HIV-1 virions and excluded from related retroviral particles including those derived from primate lentiviral lineages such as HIV-2 [166,167]. The IN-INI1 interaction has been reported to facilitate numerous steps of HIV-1 replication including transcription, particle assembly/release, reverse transcription, and integration [166,168,169,170,171,172,173]. INI1-dependent incorporation of Sin3a-associated protein 18 kD (SAP18) and histone deacetylase 1 (HDAC1) into virus particles was reported to underlie the role of INI1 in HIV-1 reverse transcription, as virions engineered to harbor catalytically inactive H141A HDAC1 were defective for vDNA synthesis [169]. Class I IN mutant W235E protein is also defective for INI1 binding, and IN mutant W235E particles accordingly lacked INI1, seemingly dispensing critical roles for the IN-INI1 interaction in HIV-1 particle production and reverse transcription [125].

#### 3.2.2. Aberrant HIV-1 Particle Morphology

Virus particles are routinely visualized by electron microscopy. Transmission electron microscopy (TEM) of negatively stained samples aided the initial characterization of HIV-1 as a retrovirus [174]. Higher resolution cryogenic electron microscopy techniques, such as cryogenic electron tomography [3,5,6,7], have further informed HIV-1 ultrastructure analyses.

The vRNP is the most electron-dense material observed via TEM [7,175]. Under these conditions, WT HIV-1 reveals a large proportion of mature particles with minor populations of eccentric and immature particles [7,53,56,58,125,176,177] (Figure 1). Class II IN mutant virus preparations by contrast harbor qualitatively reversed proportions of these respective virion populations, yielding a predominance of eccentric and immature particles [7,53,56,58,112,125,177]. Although less routinely categorized, class II IN mutant viruses can also harbor increased frequencies of particles with non-conical/malformed capsid shells [7,51] as well as particles that lack electron density altogether [51,53]. Interestingly, class I IN mutant W235E virions harbored a higher percentage of eccentric particles than the WT, though appeared more similar to the WT than did a class II IN mutant control virus [125].

Uncoating is operationally defined as the dissolution of the capsid shell during HIV-1 ingress. Initially considered a prerequisite for reverse transcription (reviewed in Ref. [178]), recent research has indicated that the capsid core remains largely intact during nuclear import and post nuclear entry, and that uncoating may occur comparatively close in space and time to integration [179,180,181,182]. We suspect that the particle morphology defects inherent to class II HIV-1 IN mutant viruses significantly influenced the prior interpretation of a role for IN in post-entry HIV-1 uncoating, which was in part based on in vitro stability assays of cores isolated from WT versus class II IN mutant C130S and IN deletion virions [157]. We would suspect that the empty and malformed cores typical of these particles to be significantly less stable than mature cores with encapsulated vRNPs.

#### 3.2.3. Defective IN Binding to vRNA

A recent study that analyzed a large panel of class II IN mutant viruses found that all of the examined IN mutant proteins were defective for binding to vRNA in vitro and in virions [56]. Consequently, the mutant viruses were defective for proper virion maturation and reverse transcription. Because eccentric particles are enriched among class I IN mutant W235E viruses [125], it will be instructive to determine the vRNA binding phenotype of W235E IN protein.

IN and vRNA are both preferentially degraded in cells following infection with class II IN mutant viruses [56,57]. We now suspect that IN and vRNA instability could have contributed significantly to prior conclusions of a role for IN in HIV-1 PIC nuclear import [153,183,184,185,186]. Recent studies have indicated that reverse transcription terminates in the nucleus only after PIC nuclear entry [179,187,188,189,190]. Thus, inherently unstable vRNA could very well confound interpretations of IN mutant viral PIC nuclear import phenotypes. By extension, we suspect that vRNA instability also underlies the reverse transcript defect that has historically been ascribed to class II IN mutant viruses. Additional research is required to assess whether secondary effects, such as IN-RT binding, might contribute to the magnitude of reverse transcription defects (see Section 3.2.1 above).

#### 3.2.4. Distinct Subcategories of Class II IN Mutant Viruses

We will leverage the comparatively recent discovery of virion IN-vRNA binding [55] as a lynchpin mediator of the class II IN mutant viral phenotype [56] to herein define a, b, and c subclasses; the different subclasses are distinguished by the phenotypic abnormalities displayed by the associated mutant IN proteins and viruses (Table 1). We herein define class IIa IN mutant viruses as those that directly compromise IN binding to vRNA without altering the functional oligomerization of the protein. Aberrant IN multimerization, which also significantly contributes to the class II IN mutant viral phenotype [56,109,111,146,191], distinguishes class IIb and class IIc mutant viruses from class IIa. Class IIb and class IIc IN mutant viruses are further distinguished based on virion protein profiles and extent of virion particle release from virus producer cells. Whereas class IIb mutant viruses contain the WT complement of virion proteins and are released normally from cells, class IIc mutant viruses contain comparatively low levels of Pol proteins and tend to release poorly from transfected cells.

Class IIa IN Mutant Viruses

This subclass comprises IN substitutions that directly impair IN binding to vRNA without affecting functional tetramerization of the protein. The abovementioned examples include substitutions of Lys264, Lys266, Arg269, and Lys273 (Figure 4) [55]. Other class IIa IN mutant viruses include K34A and R262A/R263A [56,123,151]. Further identification of class IIa mutant viruses will be aided by structural studies of IN-vRNA complexes, which are expected to elucidate additional direct protein–nucleic acid contacts.

Class IIb IN Mutant Viruses

This subclass includes amino acid substitutions that compromise functional IN tetramerization and thereby impair its ability to interact with vRNA. As mentioned above, HIV-1 IN exists in a concentration-dependent equilibrium between lower order monomer/dimer and higher order tetrameric forms. IN proteins derived from class IIb mutant viruses aberrantly multimerize in vitro, displaying enhanced formation of lower-order IN forms at the expense of IN tetramers [56]. Accordingly, these proteins were largely defective for RNA binding and unable to bridge separate RNA molecules in vitro. These data indicate that IN-RNA bridging activity contributes significantly to IN-vRNA binding in virions [56].

Class IIc IN Mutant Viruses

A significant number of HIV-1 IN missense and deletion mutant viruses have been documented to release comparatively poorly from transfected cells, which we now distinguish as class IIc IN mutant viruses [50,51,52,54,74,118,119,122,123,124,130,148,149,150,192,193,194,195,196]. Moreover, class IIc IN mutant viral particles generally harbor reduced levels of Pol proteins, such as RT and/or IN, in comparison to CA [51,52,56,84,118,150,154,196,197]. In this regard, class IIc missense mutant viruses phenocopy IN deletion mutant viruses and accordingly fail to bind vRNA due to the lack of stable IN incorporation into virions [56].

Seminal work from Bukovsky and Gottlinger [192] first revealed the connection between PR activity and the class IIc IN mutant viral phenotype. Genetic or pharmacological inactivation of PR activity countermanded the particle release defect of IN deletion mutant viruses, indicating that an intact IN domain within Gag-Pol is required to regulate Gag-Pol dimerization and hence prevent premature PR activation [84,192]. Cellular clathrin heavy and light chain proteins are specifically incorporated into divergent retroviral particles [195,196]. Clathrin packaging is mediated by short peptide recognition motifs in Gag for numerous viruses, including β- and γ-retroviruses as well as the primate lentivirus simian immunodeficiency virus from rhesus macaques (SIVmac). By contrast, mutations in RT or IN hindered the incorporation of clathrin into HIV-1 particles. Moreover, class IIc IN mutant viruses defective for particle release and Pol protein content were generally defective for clathrin incorporation [150,195,196]. Thus, clathrin incorporation into HIV-1 particles seemingly stabilizes Gag-Pol and may help to regulate Gag-Pol dimerization in the presence of active PR. Although HIV-1 particles were efficiently released from clathrin-depleted cells, the resulting virions were defective for IN protein content and were significantly less infectious than were viruses produced in the presence of normal clathrin levels [195,196]. We predict that HIV-1 produced from clathrin-depleted cells would harbor other phenotypes associated with class IIc IN mutant viruses, including aberrant particle morphology and defective reverse transcription (Table 1).

Although not as systematically documented as for proteins derived from class IIb IN mutant viruses, class IIc IN mutant proteins, when analyzed, display aberrant multimerization properties in vitro [70,109,111,146,191]. It is important to note, however, that certain distinctions between class IIb and class IIc IN mutant viral phenotypes may in part be context dependent. For example, the particle release defect of class IIc IN mutant virus particles, which is cell-type dependent, can vary in magnitude from as little as 2-fold to as much as 15-fold [50,51,52,119,192]. It therefore seems possible that some mutant viruses that reportedly release normally from transfected cells would show marginal class II IIc mutant behavior if analyzed using cell types that are comparatively nonpermissive for particle release. Akin to the discussion above on DDE active site proximal amino acid residues, different substitutions of the same IN amino acid residue can differentially affect class II IN mutant phenotypic outcome. For example, whereas mutant viruses carrying the substitution of Lys for Phe185 (F185K) show clear class IIc mutant viral behavior [56,122,124,130,195,198], F185A mutant viruses display the WT complement of virion proteins and are hence classified as IIb [54,122,195].

It is currently unclear if the extent or magnitude of IN oligomerization defect might contribute to the determination of class IIb versus class IIc IN mutant viral phenotype. HIV-1 IN can be degraded in cells via N-end rule/ubiquitination [199,200,201,202]. It therefore seems possible that mutations that may more globally impact IN oligomer integrity, such as HHCC amino acid substitutions, could enhance IN degradation during particle assembly and maturation, leading to the reduced levels of IN protein content typical of class IIc mutant viruses [56].

Class IIb and class IIc IN mutant viruses tend to harbor changes in residues that either directly partake in known IN-IN oligomerization interfaces or lie proximal to these interaction sites [56,150,191] (Figure 5B). Although the field now has several high resolution views of vDNA-bound HIV-1 IN oligomers [76,93] (Figure 3), a key limitation in determining the impact of IN structural perturbations on late events in the HIV-1 lifecycle is the lack of directly relevant structures, including IN as part of Gag-Pol or as bound to vRNA within virions. Nevertheless, most changes that perturb virus particle assembly and release map to amino acids at the IN CCD-CCD dimerization interface or residues that lie proximal to this interface (Figure 5B). These observations are fully consistent with the notion that in addition to vRNA binding, IN oligomerization is an important regulator of PR activity during HIV-1 particle assembly [150,192].

### 3.3. Complementation of Defective HIV-1 IN Mutant Virions

The infectivity defects of class I and class II HIV-1 IN mutant viruses can in large part be countermanded by expressing WT IN in trans as a fusion to the HIV-1 accessory protein Vpr (Vpr-IN_WT_) [203], which is incorporated into virion particles via a specific interaction with the p6 domain of Gag [204]. Vpr-IN_WT_ expression accordingly partially restored the reverse transcription [54] and particle morphology [7] defects of class IIc IN deletion mutant viruses. Restoration of reverse transcription was almost certainly due to supplying IN-vRNA binding function and associated mature particle formation. These data indicate that IN-vRNA binding plays an active role in HIV-1 particle maturation [7,55].

Trans-complementation systems established using purified recombinant proteins revealed that HIV-1 IN functioned as a multimer and that the NTD from one protomer within the multimer functioned in trans to the active site situated on a separate protomer [72,73]. Moreover, mutant proteins with changes in the same IN domain, such as two different DDE mutant proteins, failed to functionally trans-complement [73]. The results of these early studies are consistent with the domain-swapped organization of active IN protomers within retroviral intasome complexes (Figure 3). Extending mutant protein trans-complementation assays to Vpr-IN afforded physiological tests of the in vitro results as well as tests of virus-specific functionalities. For example, trans-expression of class IIb IN CCD mutant proteins, such as Vpr-IN_R199A_ or Vpr-IN_V165A_, efficiently complemented the infectivity defects of class I DDE catalytic triad IN mutant viruses [56,123,124,130,151,203]. Vice versa, expression of class I IN CCD mutant protein Vpr-IN_Q62K_ efficiently trans-complemented class IIb IN mutant virus V165A infection [130]. Class I IN mutant proteins carrying other CCD changes, such as Vpr-IN_K156E/K159E_ or Vpr-IN_Q62K_, expectedly failed to trans-complement class I IN DDE mutant virus, as both sets of proteins were defective for essentially the same function [124]. Proteins carrying class I mutations in separate domains, such as Vpr-IN_W235E_, by contrast efficiently trans-complemented class I IN DDE catalytic triad mutant virus [123].

These series of experiments in large part defined class I and class II as novel complementation groups that extended beyond the known HIV-1 IN domain boundaries [130] (Figure 2). In light of recent data [55,56], we can now interpret effective class I x class II IN mutant complementation as the formation of functional mixed multimers in virions, wherein the class I IN mutant protomers within the multimer supply vRNA binding function. Following reverse transcription and intasome formation, the class IIb IN mutant protomers would then provide active sites for vDNA integration. Some class II IN mutant proteins, including those with HHCC substitutions, trans-complemented active site mutant viruses comparatively poorly [56,151,203], perhaps due to IN hyper-instability, as discussed in the preceding section.

### 3.4. Class III IN Mutant Viruses

Although outside the purview of virus particle maturation, two recent studies have indicated significant roles for IN in regulating the transition from the post-catalytic strand transfer complex to transcriptionally competent provirus. Whereas the class I and class II IN mutant viruses discussed above are replication-defective, the following mutant viruses retain partial HIV-1 infectivity. Still, their study has potentially significantly expanded the multimodal functionality of IN in HIV-1 replication. For completeness, we accordingly briefly review these findings.

E212A/L213A changes within the IN CCD-CTD linker, which disrupt the interaction with cellular Ku70 [205], significantly delayed the repair of the strand transfer hemi-integrant [206]. Due to the uniqueness of the virus infection block, these authors proposed “class III” as a potential moniker for the E212A/L213A IN mutant virus [206], which we adopt herein. IN mutant virus carrying conservative lysine-to-arginine substitutions of vRNA binding residues Lys264, Lys266, and Lys274, together with the K258R change, yielded proviruses that were kinetically delayed for transcriptional activation, seemingly due to the inability for the IN mutant protein to effectively bind post-integrated LTR sequences and recruit transcriptional co-factors [207]. Although clearly acting after the completion of gap repair, we for now would also classify the tetra-Lys mutant as a class III IN mutant virus. Additional research may help to clarify whether subclassification may eventually apply to distinguish IN mutant viruses that delay the gap repair step of provirus formation versus the transcriptional competence of newly integrated proviruses (Table 1).

## 4. Allosteric IN Inhibitors

ALLINIs represent a rather diverse series of compounds that can potently inhibit HIV-1 replication in tissue culture. While INSTIs are IN active site inhibitors, ALLINIs engage the CCD dimerization interface at positions distal from the enzyme active site. The ability for ALLINIs to inhibit IN catalytic function in vitro by definition invokes an allosteric mechanism, which is why we coined the ALLINI acronym (“allosteric IN inhibitor”) for members of this drug class [208]. Other names that have been used in the literature for these compounds include LEDGIN (LEDGF/p75-IN interaction site) [209], NCINI (noncatalytic IN inhibitor) [210], and IN-LAI (IN-LEDGF allosteric inhibitor [140].

ALLINIs were discovered via two different high throughput approaches. One was based on a “wet bench” screen for IN 3′ processing activity [211]. The other, an in silico screen, searched the co-crystal structure of the HIV-1 IN CCD dimer bound to the IN binding domain of the lentiviral IN host co-factor lens epithelium-derived growth factor (LEDGF)/p75 [212] for molecules that recapitulated molecular details of the protein–protein interaction [209]. Highly similar quinoline-based compounds were discovered by both approaches which, after lead optimization, inhibited HIV-1 replication in tissue culture at low µM potency. Since these pioneering studies, ALLINIs based on several additional chemotypes, including pyridine, thiophene, and isoquinoline, have been described (see Ref. [213] and [214] for recent reviews). Prototypical quinoline and pyridine compounds are shown in Figure 6A. Recent studies have characterized several highly potent compounds based on expanded chemical scaffolds that inhibit HIV-1 replication in cell culture at effective concentration 50% (EC_50_) values of ~0.4–2.5 nM [143,215,216,217].

HIV-1 integration favors active genes [218], which is driven in part via the IN-LEDGF/p75 interaction [219,220,221]. Although initial studies concluded that the inhibition of IN-LEDGF/p75 binding during HIV-1 ingress was the principal mode of ALLINI action [209], subsequent work clarified this as a secondary outcome of drug inhibition. This line of research benefited greatly from determining ALLINI potencies during HIV-1 egress, where infectivities of viruses produced from drug-treated cells were assessed in fresh target cells, versus HIV-1 ingress, where target cells were treated with drugs at the time of infection. Such strategies revealed that ALLINIs across the board inhibited late events of HIV-1 replication more potently than early events [58,140,176,210,222,223]. Typical potency differences across studies ranged from ~10- to 100-fold, with outlier magnitudes of ~3- [176] and ~1700-fold [223] reported. The lopsided ability of pyridine ALLINI KF116 (Figure 6A) to rather specifically inhibit HIV-1 late events (EC_50_ = 30 nM) is consistent with its comparative inability to inhibit the IN-LEDGF/p75 interaction in vitro (inhibitory concentration 50% = 5 µM; [223]).

The underlying mechanism of ALLINI action is aberrant IN hyper-multimerization [111,223,224]. ALLINIs stimulate IN hyper-multimerization in vitro and in virions produced from drug-treated cells [58,111,140,176,210,222,223,224,225,226] (Figure 6B,C). Hyper-multimerized IN is defective for IN 3′ processing and strand transfer activities in vitro [140,208,211,225,227,228] and for vRNA binding in virions [55]. Accordingly, viruses made from ALLINI-treated cells harbor a predominance of eccentric particles [7,58,176,210,223,224,226,229,230,231] and are defective for reverse transcription [58,140,176,210,223] due to preferential degradation of IN and vRNA in infected target cells [57,111]. ALLINI treatment of virus producing cells does not reportedly inhibit HIV-1 particle release or polyprotein processing/virion protein content [58,176,210,222,223,226]. Thus, ALLINI-treated viruses phenocopy HIV-1 IN class IIb mutant viruses (Table 1).

**Figure 6 viruses-14-00926-f006:**
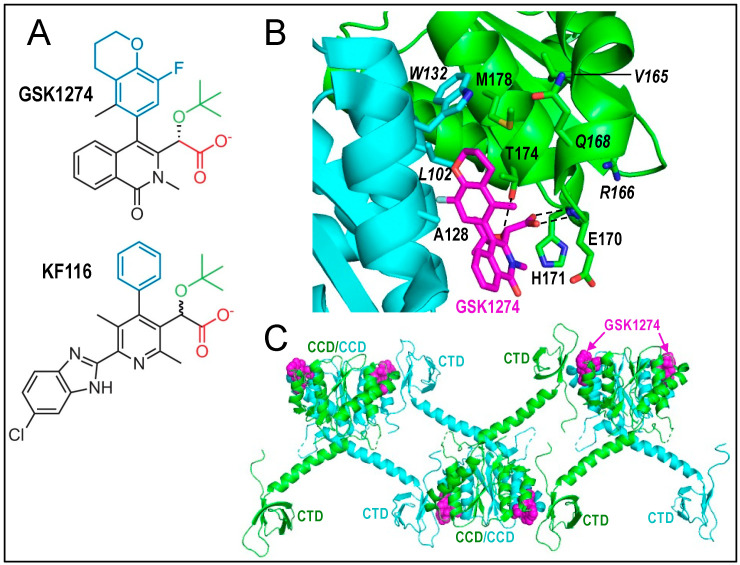
Representative ALLINI chemotypes and IN binding modes. (**A**) Structures of quinoline GSK1274 [222] and pyridine KF116 [223] compounds. See text for color descriptors. (**B**) X-ray crystal structure of GSK1274 bound to the IN CCD dimer (PDB accession code 4OJR [222]); the IN protein harbored the solubility-enhancing F185K substitution [74,232]. CCD dimerization interface residues in close contact to GSK1264 are shown. Nearby Val165 and Arg166 are additionally highlighted. Amino acid residues that when mutated yield class IIb (Val165, Gln168) or class IIc (Leu102, Trp132, Arg166) IN mutant viruses are labeled in italics. Atoms are colored as follows: red, oxygen; blue, nitrogen; mustard, sulfur; and light blue, fluorine. Dashed lines, hydrogen bond interactions. (**C**) Structure depicting ALLINI-induced IN hyper-multimerization (PDB accession code 5HOT [230]). The full-length IN construct harbored Y15A and F185H amino acid substitutions. The IN NTD was not resolved in the electron density map; other domains are indicated. GSK1264 is colored magenta in panels B and C.

Despite fairly large structural divergence among compounds, ALLINIs harbor three commonalities, including a carboxylic acid (red in Figure 6A), large hydrophobic (blue), and compact aliphatic (typically tert-butoxy; green) moiety emanating from central pharmacophores. The hydrophobic side chain buries into the CCD-CCD binding pocket, where it contacts hydrophobic amino acids from both monomers of the IN dimer. The aliphatic moiety helps to fill the base of the binding pocket, while the carboxylic acid interacts with the backbone amides of IN residues Glu170 and His171 of the green IN protomer (Figure 6B). This latter interaction recapitulates the binding mode of LEDGF/p75 hotspot residue Asp366 [212,233], explaining why many ALLINIs are effective competitors of the IN-LEDGF/p75 interaction [140,209,222,226,227].

IN hyper-multimerization is templated via higher-order intermolecular interactions between CCD dimer-bound compound and the CTD of an adjoining IN multimer [111,230,234,235] (Figure 6C). Substitutions that confer ALLINI resistance in cell culture generally occur at CCD-CCD binding interface residues [209,215,223,226,227,230,236], though CTD-proximal changes can also confer resistance [230]. The CCD H171T change conferred similar ~40-fold resistance to quinoline ALLINI BI-D when incorporated into Gag-Pol or expressed in trans as Vpr-IN_H171T_, indicating that IN, as opposed to Gag-Pol, is the primary target of ALLINI action [58].

Analyzing IN mutant viruses that arose during the selection of resistance to ALLINI KF116 uncovered a striking example of PR-IN interplay during the late events of HIV-1 replication [197,223]. Robust resistance required three amino acid changes in and around the CCD dimerization interface, T124N/V165I/T174I, which accumulated in successive fashion. While the initial T124N change imparted marginal fitness cost, T124N/T174I was practically a dead virus (~0.14% of WT HIV-1 infectivity). T124N/T174I mutant virions were predominantly immature due to lack of effective PR activity, which practically eliminated proteolyzed Gag and Gag-Pol products from the virus particles [197]. The subsequent V165I mutation in large part counteracted the deleterious effects of the T124N/T174I changes, with T124N/V165I/T174I IN mutant virus displaying 17% of WT infectivity and ~30% to 70% of WT levels of processed Gag and Gag-Pol products. These observations indicate that the intimate interplay between IN and PR during HIV-1 maturation could potentially be exploited for novel antiretroviral inhibitor development.

## 5. Conclusions and Perspectives

Recent findings have significantly informed the multimodal functionalities of HIV-1 integrase. IN seemingly plays an important role to regulate PR activation during the initiation of HIV-1 maturation. All three Pol protein domains, PR [237], RT [238], and IN [72,73,76,80], function as obligate enzyme multimers, and the IN domain, possibly in concert with cellular clathrin, seems to regulate Pol/PR dimerization, a prerequisite for PR activation. Gag-Pol proteins lacking the IN domain or harboring changes that grossly impact IN oligomerization lack this regulatory capacity, leading to defects in virion protein composition and particle yield from virus producing cells (the class IIc IN mutant viral phenotype). Although currently unknown, we would suspect that the IN domain functions as a dimer in this capacity. Based on results using purified IN proteins, which demonstrated tetramer-specific RNA binding activity [56], we would by extension doubt that the IN domain of Gag-Pol could effectively bind vRNA, although surely this could be directly tested.

Although the timing and order of post Gag-Pol cleavage events are poorly understood, we propose that IN tetramerizes soon after its liberation from Gag-Pol, possibly aided by vRNA binding. IN prebound by nucleic acid is recalcitrant to ALLINI-induced hyper-multimerization in vitro [208,227], and ALLINIs fail to impact the infectivity of cell-free HIV-1 particles [58], indicating that essentially all functionally relevant IN molecules are sequestered as part of the vRNP in virions. Inhibition of IN-vRNA binding through ALLINI-induced IN hyper-multimerization, or via amino acid substitutions that significantly reduce IN tetramer formation (class IIb IN mutations) or directly alter vRNA binding (class IIa IN mutations), yield eccentric particles that are defective for reverse transcription and HIV-1 infection.

Despite these advances, several unanswered questions remain about the role of IN-vRNA binding in vRNP encapsulation. We can envision different models to account for this outcome, none of which are mutually exclusive. One model invokes a direct interaction between IN and CA. Although IN and CA can be seen to covary genetically [239], to date we are unaware of evidence for a direct protein interaction. An extension of this model would invoke an interaction between IN and a Gag processing intermediate, again for which there is currently no direct evidence. Other models that do not invoke direct interactions of IN with CA/Gag can also be envisaged. NC’s RNA binding and chaperone activity helps to arrange vRNA into thermodynamically stable conformations within the vRNP [240]. If the kinetics of NC chaperone activity is significantly slower than IN-vRNA binding/bridging activity, loss of the “faster” activity could potentially result in eccentrically located, NC-containing vRNPs. Kinetic tests of NC and IN RNA binding and bridging/condensation activities would seem readily approachable.

Structural determinants for specific, high-affinity binding of IN to cognate sites on vRNA are unknown. CLIP-Seq results coupled with bioinformatic tools could be employed to identify conserved RNA sequence and/or structural features that are preferentially targeted by IN in virions. Complementary mass spectrometry-based footprinting and site-directed mutagenesis of IN pointed to several basic amino acids in the CTD and proximal tail region that likely engage in electrostatic interactions with vRNA [55,56]. However, elucidation of specificity determinants for IN-vRNA interactions will likely have to await structural studies. Furthermore, comparative analysis of a future high-resolution IN-vRNA structure with available full-length IN-ALLINI structures [230,235] could help us to better understand the underlying mechanisms for ALLINI action. For example, do these inhibitors block IN-vRNA binding by directly shielding nucleic acid binding sites on IN, or indirectly, via inducing IN hyper-multimerization?

Another unanswered question is the extent to which IN-vRNA binding plays roles in the replication and particle maturation processes of other retroviruses, although some clues can be gleaned from prior analyses of α-retroviral IN proteins and γ-retroviral Moloney murine leukemia virus (Mo-MLV) IN mutant viruses. As evidenced by tight binding to RNA-conjugated chromatography columns such as Poly(U)-Sepharose/agarose, virion-derived as well as recombinantly produced α-retroviral IN proteins display affinity for RNA [35,241,242]. Indeed, the β portion of the RT α/β heterodimer was shown to specifically bind the tRNA^Trp^ primer required for reverse transcription prior to knowing that this region of RT-β was the viral IN [243,244]. The binding of recombinant Rous sarcoma virus IN to RNA in vitro moreover mapped to the CTD [242], which is the HIV-1 IN domain that harbors the predominance of mapped RNA binding residues [55,56]. Partial or full deletion of Mo-MLV IN resulted in ~10-fold reductions in reverse transcription [245,246], which is reminiscent of the HIV-1 IN class II IN mutant phenotype. Moreover, trans-incorporation of WT IN, in this case as a Gag-IN fusion protein, restored reverse transcription to Mo-MLV IN deletion mutant viruses [246]. Additional work that comprehensively addresses the roles of IN-vRNA binding in virus particle maturation and reverse transcription across a variety of viruses would more fully inform the non-catalytic roles of IN in retroviral replication.

## Figures and Tables

**Figure 1 viruses-14-00926-f001:**
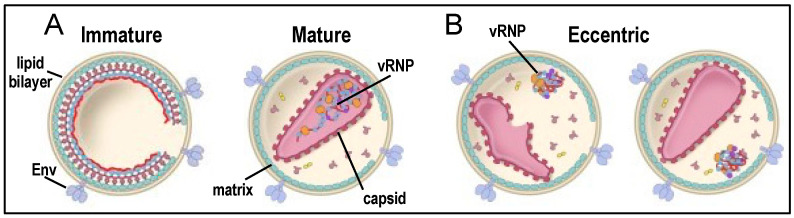
HIV-1 virion particles. (**A**) Schematic of immature and mature virions. Note the vRNP encased in the capsid shell in mature particles. (**B**) Eccentric virions with deformed (left) or closed (right) capsid. Env, envelope glycoproteins. See main text for additional descriptions. Adapted from Ref. [7].

**Figure 2 viruses-14-00926-f002:**
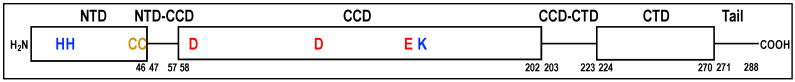
HIV-1 IN domains and phylogenetically conserved amino acid residues. Upper labels denote domains, interdomain linkers, and C-terminal tail region. Active site DDE residues are colored red, nitrogen-containing side chains blue, and cysteines are colored dark yellow. CCD residue Lys159, which is conserved among retroviral but not retrotransposon INs, interacts with the vDNA end [75,76]. Numbers, based on Refs. [77,78], denote domain and linker boundaries.

**Figure 3 viruses-14-00926-f003:**
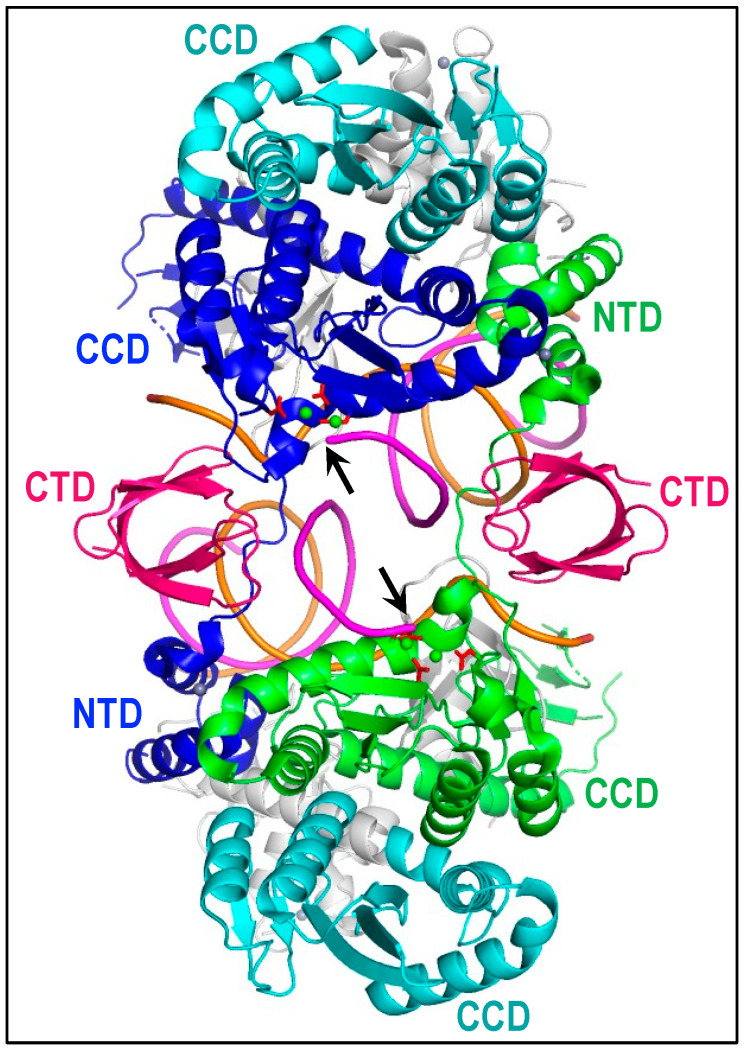
HIV-1 intasome structure (protein database (PDB) accession code 6PUT). Cyan-blue (top) and green-cyan (bottom) IN dimers interact via the CCD-CCD dimerization interface. The internal blue-green IN dimer is bound to vDNA, which is colored orange and magenta to highlight non-transferred and transferred DNA strands, respectively. The 3′ ends of the magenta transferred strands, which are indicated by arrows, are engaged by inner IN dimer DDE active sites (red sticks in association with two calcium ions, which are shown as green spheres). NTD-associated grey spheres are zinc. Synaptic CTDs are in hotpink to indicate their origin from peripheral IN tetramers, which were otherwise omitted from the atomic model. Gray is used to deemphasize secondary structural elements that do not directly contribute to the CIC.

**Figure 4 viruses-14-00926-f004:**
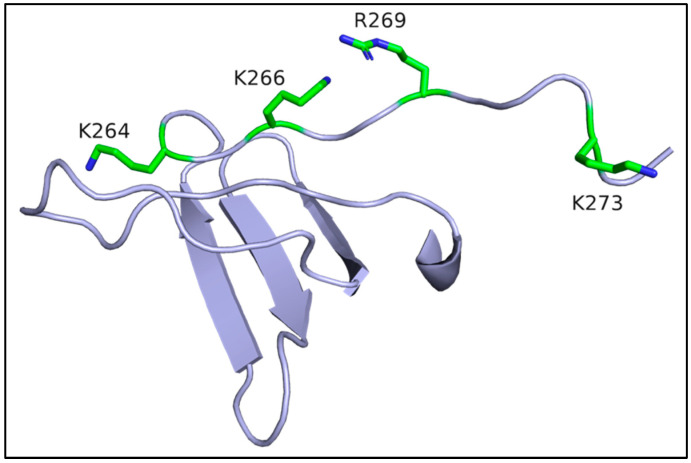
Cartoon representation of the CTD and six downstream residues (amino acids 271–276) of the IN tail region (PDB accession code 5HOT). The side chains of IN residues Lys264, Lys266, Arg269, and Lys273, which are implicated in vRNA binding, are shown as sticks.

**Figure 5 viruses-14-00926-f005:**
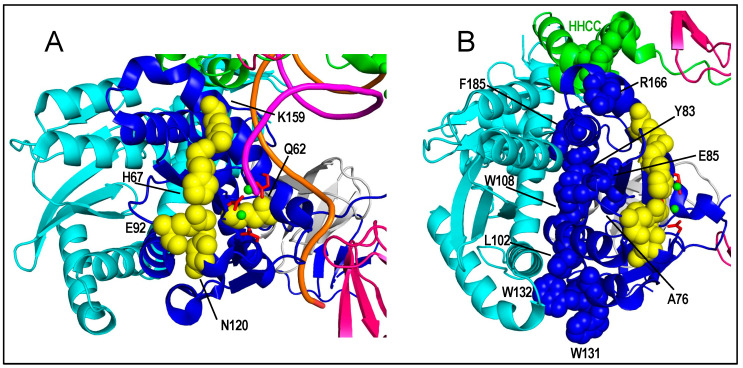
Class I and class II IN mutations and IN oligomerization interface proximity. (**A**) The blue CCD from the HIV-1 intasome in Figure 3 was resected and positioned to highlight active site proximal amino acid residues (in yellow space fill) that when changed yield the class I IN mutant viral phenotype. DDE active site residues are shown as red sticks. (**B**) The panel A image was rotated clockwise ~45° around the y-axis to highlight the cyan CCD-blue CCD dimerization interface. Blue CCD and green NTD amino acid residues that when mutated yield the class IIc IN mutant viral phenotype (see Table 1, below) are shown in space fill [51,56,122,124,148,149,150]; His12, His16, and Cys40 are collectively labeled HHCC. The panel B image highlights oligomerization interface proximity of residues that when changed yield class II IN mutant viruses. For clarity, the DNA duplex was omitted from panel B.

**Table 1 viruses-14-00926-t001:** Phenotypic classes of HIV-1 IN mutant viruses.

Mutant Virus Class	Associated Replication Defect (s) ^1^
Class I	Integration
Class IIa	PM, IN-vRNA binding, rt, integration
Class IIb	PM, abbIN, IN-vRNA binding, rt, integration
Class IIc	VR, PM, Pol content, abbIN, IN-vRNA binding, rt, integration
Class III	Post-integration DNA repair/transcription

^1^ PM, particle morphology; rt, reverse transcription; abbIN, aberrant IN multimerization; VR, virus particle release from transfected cells.

## Data Availability

Not applicable.

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
