# Peer review of "Multimodal Functionalities of HIV-1 Integrase"

_viruses, 2022, doi:10.3390/v14050926_

Round 1

Reviewer 1 Report

The authors present a very interesting and timely review of HIV-1 integrase and virus particle maturation.  The HIV-1 story is very well covered and clearly presented.  They present a well-written article demonstrating a command of the HIV-1 IN field. The emphasis on association of IN mutants to the various described Classes is highly appropriate.  The classification criteria in Table 1 is interesting and should be useful in the future. They suggest that additional studies are needed to address the roles of IN-vRNA binding in virus particles maturation and reverse transcription across a variety of retroviruses.  The authors cited several MLV IN studies. There has been a significant number of earlier α-retrovirus IN studies that focus on 1) IN in ribonucleoprotein particles (RNP) derived from virus core particles, 2) the IN moiety on the β subunit of the αβ reverse transcriptase and its role in specifically binding primer tRNATrp, 3) the ability of IN to bind different ssRNA polymers in vitro, 4) mapping the domains of IN necessary for binding ssRNA by IN truncations, 5) IN mutants affecting gag-pol processing in virions while not affecting integration, and 6) three adjacent aptamer ssRNA (4 bases) binding pockets identified on the crystal structure of dimeric IN core (1.8 Å resolution) . Citing several of these α-retrovirus studies would be helpful to investigators as a guide for future studies on IN-RNA binding properties of different retroviruses.

Minor Points:

P1, maybe insert Fig. 1A after [3] on line 41. There is significant discussion of the virus structure thereafter without visual reference. It would be helpful to readers not familiar with the retrovirus structure.

P 3, Fig. 2 is too large to fit in the lines.

P 5, Fig. 3. The gray spheres identifying zinc are not apparent in the NTD.

P 8, Fig. 5.  It is very difficult to visualize the amino acid blue spheres with the background blue ribbon structures in both Fig. 5 A and B. The yellow spheres in 5B are excellent.  A change of color with the background or spheres would be important to better visualize and understand the Class I and II IN mutants. The different classes of IN mutant viruses makes sense.

Articles on α-retrovirus IN-RNA binding mentioned above:

1) IN in ribonucleoprotein complexes derived from virus core particles—J. Virology (1974) 13: 513-528 and Biochem. Biophys. Acta (1974) 361:53-58.

2) IN moiety on the β subunit of αβ reverse transcriptase and its role in binding the specific primer tRNATrp --PNAS (1975) 72: 2535-2539 and Virology (1976) 75: 26-32.

3) ability of IN to bind ssRNA -- Virology (1978) 89: 119-132; J. Virology (1988) 62: 2358-2365;

4) mapping ssRNA binding domain to C-terminus on truncated IN in vitro, J. Virology (1991) 65:1160-1167; point and deletion mutagenesis of IN (from S262 to C-terminus end) for single round infectivities —Communications Biol. (2021) 4:330.

5) IN mutants affecting gag-pol processing in virions—Virology (1984) 137:358-370 (Possibly a Class IIc mutant?).

6) three adjacent pockets identified on the crystal structure of IN dimer core (1.8 Å resolution) that have docked aptamer ssRNA (4 bases)--PLoS One (2011) 6: e23032.
